# Higher-Order Markov Model-Based Analysis of Reinforcement Learning in 6G Mobile Retrial Queueing Systems

**DOI:** 10.3390/s25237245

**Published:** 2025-11-27

**Authors:** Djamila Talbi, Zoltan Gal

**Affiliations:** Faculty of Informatics, University of Debrecen, Egyetem ter 1, 4032 Debrecen, Hungary

**Keywords:** retrial queueing system, deep Q-network reinforcement learning, B5G/6G, Markov chain, higher-order Markov chain, spectral gap, queueing theory, dynamic time warping

## Abstract

The dynamic behavior of the retrial queueing system following the incorporation of Deep Q-Network Reinforcement Learning in 6G mobile communication services is examined in this study. The proposed method lies in analyzing the DQN-RL agent’s learning convergence by using the first- and second-order Markov chain method. By simulating the temporal evolution of reward sequences as Markov and second-order Markov chains, we can quantify convergence characteristics through mixing time analysis. To capture a wide operational landscape, a thorough simulation framework with 120 independent parameter combinations is created. The obtained results indicate that Markov chain analysis confirms 10 training episodes are more than sufficient for policy convergence, and in some cases, as few as 5 episodes allow the agent to enhance the mobile network performance while maintaining low energy consumption. To assess learning stability and system responsiveness, the mixing time of DQN RL rewards is calculated for every episode and configuration. A deeper understanding of the temporal dependencies in the reward process can be gained by incorporating higher-order Markov models. This paper concentrates on studying the learning convergence using an analysis of the Markov model’s spectral gap properties as an indicator. The results provide a rigorous foundation for optimizing 6G queueing strategies under uncertainty by highlighting the sensitivity of DQN convergence to system parameters and retrial dynamics.

## 1. Introduction

### 1.1. Context and Motivation

A new communication paradigm is introduced by the sixth generation (6G) wireless networks, including ultra-dense connectivity, low-latency requirements, and intelligent resource management. It is expected to support highly dynamic, heterogeneous services, real-time control, autonomous communication, and large-scale sensor deployments. Achieving such requirements creates substantial challenges for resource allocation, congestion control, and service coordination, particularly under fluctuating network conditions and bursty traffic loads [1].

For decades, queueing theory has served as a fundamental analytical framework to evaluate network performance. Increasing complexity and uncertainty of the 6G environment demand adaptive and self-optimizing mechanisms beyond conventional static queueing analysis. Therefore, integration decision-making-based learning would offer a promising direction for achieving adaptive, data-driven control of service process and network resources. Our investigation focuses on the learning dynamics of the integrated system, particularly concentrating on the policy stabilization importance in the quality of the training.

### 1.2. Research Gap

In the literature, several studies have examined Markovian queueing systems and Reinforcement Learning (RL) based optimization for network management [2,3,4]. The researchers in [2] propose a Deep Reinforcement Learning active queue management scheme at fog/edge nodes, using a DQN-based controller in order to balance delay and throughput during congestion. Paper [3] surveys how graph neural networks and deep reinforcement learning enhance end-to-end networking tasks, while [4] analyzes Markovian retrial queueing models for cognitive radio networks, focusing on sensing behavior and priority interactions between different users. These works demonstrate the advantage of RL and Markovian modeling in networking by addressing congestion control, E2E resource optimization, or spectrum-access processes rather than RL-based queue-level decision making. This distinction underscores the need for a hybrid framework that captures Markovian queue behavior and adaptive learning for optimal control under various network conditions. Based on our investigation, most existing models are limited in one or more aspects. The classical M/M/1 or M/M/1/K queueing models often assume fixed parameters and lack adaptability to real-traffic variations. Additionally, current machine learning RL applications in networking, and mainly in 6G, primarily focus on routing, spectrum access, or power allocation. Few existing research papers have addressed the integration of RL with retrial queueing systems to dynamically balance service efficiency, queue stability, and fairness under different network conditions. Notably, this gap highlights the need for a hybrid modeling framework capable of capturing both stochastic queueing dynamics and a smart adaptive learning system. Such an approach would allow the system to learn optimal service strategies while maintaining analytical tractability through Markovian representation.

### 1.3. Contribution of the Research Study

The presented study proposes an integrated 6G-based retrial queueing system with deep N-network reinforcement learning. This provides a novel methodology for adaptive queue management and service optimization. The main contribution of the paper is summarized as shown in the workflow in Figure 1 and as follows:Deep Q-Network Reinforcement Learning: We choose to use the deep Q-network RL as a real-time decision-making neural network. This will serve as the native intelligent box in the 6G communication environment.M/M/1/K Retrial Queueing System: We model a queueing system in a 6G environment to simulate multiple scenarios under varied conditions. The main aim is to evaluate and study the behavior of such systems.Extracted Agent Reward for Analysis Framework: After the DQN-RL integration, we explore the influence of the rewards on the overall study. Moreover, we study the effect of the tunable parameters used in the reward function on the quality of the Agent teaching.Higher-order Modeling and interpretation: We incorporate the first and second-order Markov models to provide a deeper understanding of the system using the agent’s rewards. Investigating the behavior of the first and second spectral gaps helps us to deeply interpret the findings. At the same time, Dynamic Time Warping (DTW) is used to calculate the similarity level between the steady states.Learning Convergence Point and Optimal Training Episode: By passing the previous steps, we can easily capture the optimum number of episodes needed to reach a learning convergence point and prevent over-teaching our RL model.

This paper is structured as follows: Section two provides an overview of the related research papers, summarizing existing approaches to the topic and the methods used. Section three presents the theoretical background and modeling framework. Section four details the presented methodology, including several aspects of the integration process into the simulation environment. Section five discusses the experimental results and performance. Finally, section six concludes the paper with key findings, limitations, and potential directions for future work within 6G systems.

## 2. Related Work

This section reviews the main contributions of the recent studies by other researchers related to our research topic. It is separated into three main subsections. Each subsection groups one main approach.

### 2.1. Queueing Systems in Wireless Networks

In the context of wireless networks, queue management is addressed through many perspectives. 5G/6G core networks are targeted in [5], introducing a dynamic application-layer queue management system for supporting dynamic Quality of Service (QoS) and programmable network slices. Cognitive radio channels are studied in [6], modeling the behavior of secondary users that opportunistically access channels allocated to primary users. A priority-based queue management method called CERED has been proposed by researchers in [7] concerning congestion in wireless sensor networks. The study’s method gives high priority to real-time packets and optimizes queue handling to reduce delays while maintaining system efficiency.

Queue management was also discussed in large and complex models. Machine learning is used in [8] to approximate queue behavior in finite-buffer systems, offering a scalable alternative to the traditional models. From an industrial IoT network perspective, ref. [9] proposes a distributed algorithm and multi-hop strategies to maintain queue stability, ensuring network-wide throughput and energy efficiency.

In specialized wireless network scenarios, a finite-capacity queue with server vacations is modeled for energy-harvesting wireless sensor nodes [10], while in ref. [11], a single-server queue is modeled with multiple service types and interruptions. A software-defined UAV networking framework is proposed by researchers in [12] for dynamic and large-scale wireless networks.

Across these papers, the main results show that advanced queue management can significantly improve network performance by reducing delays, stabilizing queues, and optimizing resource use in diverse wireless network scenarios.

### 2.2. Reinforcement Learning in Networking

For exploring the role of reinforcement learning and AI in enhancing wireless networks, the surveys in paper [13] include machine learning and deep learning to enable intelligent decision-making, automation, and adaptive management in next-generation wireless networks, while paper [14] uses multi-agent reinforcement learning (MARL) in distributed 6G networks to examine how decentralized, collaborative learning can optimize the performance. An RL-based resource allocation method is used for vehicular cognitive radio networks in paper [15]. The study demonstrates improvements in throughput, latency, and energy efficiency in highly dynamic vehicular environments.

RL is widely used for optimization purposes in 6G and beyond wireless networks. Paper [16] proposes a blockchain-enabled 6G wireless sensor network framework, combining RL and optimization algorithms to enhance security, throughput, and packet delivery. Paper [17] introduces an RL-based channel access mechanism for massive IoT deployments in next-generation WI FI networks.

For 5G/6G networks and beyond, network slice mobility is addressed in ref. [18]. The authors propose a prediction-based federated deep RL framework to anticipate system changes, optimize slice migration, and improve long-term performance while reducing communication overhead. Researchers in [19] introduce a hypergraph-based interference model and an RL algorithm to manage spectrum resources and maximize network throughput in dense device deployments.

These papers prove that RL-based techniques enhance wireless network performance throughout all the key performance indicators, ensuring efficient resource use and supporting adaptive, intelligent decision-making in dynamic and large-scale environments.

### 2.3. Markov Models in Learning Analysis

In this part, we focus on the related papers that focus on applying Markov models to analyze and optimize different networked systems. An 8-state discrete-time Markov chain is used in [20] to model primary user activity in cognitive radio networks. Using a special decision algorithm to improve channel utilization. A cellular automata Markov chain is combined with neural networks to predict agricultural condition changes in the Amazon [21]. A 3D Markov chain model is developed in [22] to evaluate IEEE 802.11 MAC performance in UAV ad hoc networks, quantifying the key performance indicators under high-mobility network conditions.

A Markov chain is also used to capture behaviors in wireless systems. Therefore, researchers in [23] use a Markov-based approach to infer network topology from limited observations. In [24], Markov modeling is applied for trust management in wireless sensor networks, improving security, and enabling rapid detection of malicious nodes. The same idea was used for energy harvesting in cell-free MIMO networks, using a Markov chain to track energy consumption and optimize network performance.

Overall, these studies demonstrate that Markov chain models provide a powerful tool for analyzing dynamic behaviors and optimizing performance in wireless networks and complex systems.

## 3. System Model and Problem Formulation

In this section, we present the modeling framework for the proposed RL-RQS model. We describe the retrial queueing system using the capture of realistic 6G network behavior. We define reinforcement learning and higher-order Markov modeling to optimize decision-making under dynamic traffic conditions.

### 3.1. Retrial Queueing System Model

Among the existing queueing models, we chose a retrial queueing system. It is a system where customers who find the server busy upon arrival do not leave permanently but instead join a finite queue. Retrial queues are particularly relevant for wireless networks and telecommunication systems, where the packets or connection requests may need multiple attempts due to various scenarios. These scenarios may include unavailability of the channel, server congestion, or inference. These kinds of systems capture realistic behaviors in networks where blocking or immediate service denial is common.

Queueing systems are represented using Kendall’s notation: A/B/C/K/N/D. Each character refers to a specific property of the system:

A: Arrival process (e.g., M = Markovian/Poisson arrivals)B: Service time distribution (e.g., M = exponential)C: Number of serversK: Queue length capacity (maximum number of customers that wait in the queue)N: Population size (default ∞)D: Queue discipline (e.g., FCFS: First Come First Served)

For retrial queues, the classical Kendall’s notation is extended to include the *orbit*, capturing the behavior of customers who retry after an initial failure to join the queue.

In order to model the 6G core network scenario, we find that the M/M/1/K retrial queue system is particularly suitable for the following reasons:

M/M/1 refers to a single server with Poisson arrival (λ) and exponentially distributed service time (μ). This is common in network traffic analysis [25,26,27,28].K defines the finite capacity of the queue. This aligns with the practical limits of 6G network buffers or slices in certain scenarios.Retrial mechanism of the system, which captures the repeated attempts of packets that fail to access the server, reflecting dynamic QoS requirements and congestion in 6G scenarios.

The system explained provides a tractable framework. It helps to study multiple KPIs in scenarios with limited resources and high variability in traffic.

The server serves the clients with an exponential service time of mean 1/μ; therefore, the probability of a service completion in dt is:(1)$$P_{service}\approx \mu dt$$

The exponential distribution allows the system to be modeled as a Continuous-Time Markov Chain (CTMC). The state transitions depend on the rates λ, μ, and the retrial rate θ. The probability of a new customer arriving in a small dt is approximately:(2)$$P_{arrival}\approx \lambda dt$$

The clients that could not join the queue join an orbit and retry after an exponential delay with mean 1/θ. The probability of retrying within dt is:(3)$$P_{retry}\approx \theta dt$$

We define the balance equation for state probabilities as follows:(4)$$\frac{dP_{n}}{dt}=\lambda P_{n-1}-\left(\lambda +\mu \right)P_{n}+\mu P_{n+1},\;\;\;1\leq n<K$$

For a full system (n=K), retrials are included:(5)$$\mu P_{K}=\lambda P_{K-1}+\theta P_{orbit}$$
where the orbit probability is:(6)$$P_{orbit}=\frac{\lambda P_{K}}{\theta }$$

The system is considered stable as long as the traffic intensity satisfies:(7)$$\rho =\frac{\lambda }{\mu }<1$$

Satisfying this condition, the system remains stable enough that arrivals do not exceed the service capacity.

### 3.2. Deep Q-Network Reinforcement Learning Framework

Among the types of machine learning that are used for enhancing the performance of any system, we find Reinforcement Learning. RL is a decision-making paradigm where the agent interacts with the environment by observing states, executing actions, and receiving rewards as a type of feedback. By doing so, the agent aims to learn the optimal policy that maximizes the expected cumulative reward over the learning time. Q-learning is a method where the agent estimates the state-action value function Q(s,a), making a prediction for the long-term utility of performing an action a in state s. Traditional Q-learning is a tabular representation of Q values. It becomes computationally infeasible in large or continuous state spaces, particularly for next-generation wireless network scenarios.

This limitation has been addressed by proposing Deep Q-Networks. DGN employs deep neural networks to approximate the Q-function. Thus, it helps the agent to generalize across high-dimensional inputs and make decisions in complex environments.

The DQN framework is characterized by several components, as explained in Table 1, and has the shown structure in Figure 2. The primary objective of the training is to minimize the temporal-difference (TD) error. Which is defined by the loss function as follows:(8)$$L\left(\theta \right)=E_{\left(s,a,r,s\prime \right)}[{(r+\gamma \underset{a^{\prime }}{\max}Q(s^{\prime },a^{\prime };\theta^{-})-Q(s,a;\theta ))}^{2}]$$
where the symbols represent the following:
s and $s^{\prime }$ denotes the current and the next states.a is the action taken in state s, and $a^{\prime }$ represents possible next actions.r is the immediate reward.γ∈[0,1] is the discount factor.Q(s, a;θ) is the approximated action-value function.$Q\left(s^{\prime },a^{\prime };\theta^{-}\right)$ is the target network used to stabilize learning.

This formulation ensures that the DQN updates its parameters by reducing the squared Temporal-difference error between the predicted and target action values.

Over time, and by refining the Q-function, the DQN framework enables the agent to approach near-optimal policies, even in complex and highly dynamic environments. Therefore, the DQN framework has become a promising tool for decision making, resource allocation, and scheduling optimization for many complex systems such as 6G communication systems, where we find that the network conditions exhibit stochastic and time-varying behavior.

### 3.3. Markov and Higher-Order Modeling

A Markov chain is used for characterizing the stochastic behavior of dynamic systems, particularly in next-generation wireless networks, where the system states evolve over time. First- and higher-order Markov chains have been widely applied in network modeling and queueing systems to predict state transitions and system dynamics under uncertainty. The first-order model assumes memoryless transitions, while second-order models account for dependencies on previous state, providing more accurate predictions in environments with temporal correlations. As it was explained in previous sections, prior studies rarely integrate these models with deep reinforcement learning for adaptive resource allocation in 6G mobile networks, which is the focus of this work.

Following the first-order Markov model, the future state of a process depends on its present state (see Figure 3a). This property benefits memoryless transitions between network states. A first-order Markov chain is a stochastic process ${\left\{X_{t}\right\}}_{t\geq 0}$ with Markov property:(9)$$P\left(X_{t+1}|X_{t},\;X_{t-1},\ldots ,X_{0}\right)=P\left(X_{t+1}|X_{t}\right)$$

The state space is $S=\{s_{1},s_{2},\ldots ,s_{n}\}$, for all t, and the transition are governed by the transition tensor P:(10)$$P_{ij}=P\left(X_{t+1}=s_{j}|X_{t}=s_{i}\right),\;\;i,j\in S$$

For stationary distributions, π satisfies the following condition:(11)$$\pi P=\pi ,\;\;\sum_{i}\pi_{i}=1$$

The mixing rate is given by the spectral gap:(12)$$1-\left|\lambda_{2}\right|$$
where $\lambda_{2}$ is the second-largest eigenvalue of P. The hitting time is inversely related to the spectral gap:(13)$$T_{hit}\approx \frac{1}{1-\left|\lambda_{2}\right|}$$

A large spectral gap means a small hitting time, indicating rapid state transitions and good mixing.

Real-world 6G environment often exhibits dependencies that extend beyond immediate past states, leading to the use of higher-order Markov models. Due to the considered past events, these models capture more complex temporal correlations, providing a more comprehensive representation of system dynamics and performance fluctuations. The second-order chain relaxes the memoryless property to two past steps, as shown in Figure 3b and presented as follows:(14)$$P\left(X_{t+1}=x_{t+1}|X_{t}=x_{t},\;X_{t-1}=x_{t-1},\ldots \right)=P\left(X_{t+1}=x_{t+1}|X_{t}=x_{t},\;X_{t-1}=x_{t-1}\right)$$

The second-order transition tensor is defined as:(15)$$P_{ijk}=P\left(X_{t+1}=s_{k}|X_{t}=s_{j},X_{t-1}=s_{i}\right),\;\;i,j,k\in S$$

Such models enable accurate prediction and decision-making under uncertainty, supporting adaptive resource allocation and reliability estimation, particularly for high-speed communication networks. The integration of the first- and second-order Markov models with DQN-RL NN further enhances the system’s decision-making to learn and adapt to dynamic environment patterns.(16)$${SG}_{1}=1-\lambda_{2}$$(17)$${SG}_{2}=1-\lambda_{3}$$

${SG}_{1}$ represents global inter-cluster spectral pap, and ${SG}_{2}$ gives the local intra-cluster spectral gap. Common interpretation of the (${SG}_{1}$, ${SG}_{2}$) metric pairs is given in Table 2.

The first- and second-order spectral gaps (${SG}_{1}$, ${SG}_{2}$) provide valuable insight into the system’s temporal dynamics and stability. In the context of 6G networks, these analyses further facilitate the identification of metastable communication states and dynamic bottlenecks that are expected to be one of the challenges of the integration of many technologies. We believe that applying methods such as the first- and second-order Markov models with NN-based reinforcement learning would help the system capture temporal dependencies, resulting in enhanced convergence, diminished uncertainty, and more intelligent adaptation to environmental fluctuations.

## 4. Methodology

The proposed method aims to model and optimize the upcoming wireless communication access by integrating the DQN-RL into the RQS framework. This hybrid approach is expected to capture both the stochastic dynamics of user access requests and the adaptive decision-making required to maintain system stability under various traffic conditions. The access process between the Mobile Terminals (MTs) and the Access Point (AP) is modeled using/M/1/K retrial queue (see Figure 4) following Kendall’s notation. The arrival rates is symbolized with λ, serving rate by μ, and queue size by K.

The retrial mechanism mirrors the practical behavior of 6G terminals. This end, attempt re-access after failed initial connections caused by temporary congestion, beam misalignment, or fading.

In order to introduce adaptive intelligence to the queueing system to match the native intelligence nature of the 6G, we chose to integrate an RL mechanism into the RQS framework. This enables adaptive intelligence in the queueing systems, enabling them to learn optimal control strategies from experience rather than relying on static probabilistic assumptions. The RL approach is chosen because 6G access is highly dynamic and stochastic, making static policies insufficient, while DQN allows the agent to learn optimal actions from interaction, balancing queue stability and throughput in real time.

In the proposed model, the agent acts as an upper-layer controller that continuously observes the system’s state, represented by the current queue size and AP status (busy or idle) (see Figure 5 and Algorithm 1), and selects optimal actions to maintain service efficiency. The agent can choose one action from the actions list that contains two choices:Action 1: No intervention, the system operates normally.Action 2: Immediate interaction to force the AP to serve one queued MT.

In Figure 5, $O_{t|t+1}$ represents the observation at time t associated with the transition to time t+1. $R_{t|t+1}$ represents the reward obtained at time t as a result of the transition from t to t+1.

The agent’s goal is to minimize queue congestion and maximize service throughput, which is formulated by the reward:(18)Reward=α·Δ(served MTs)−β·Δ(queued MTs)
where Δ(served MTs) and Δ(queued MTs) are the changes in the number of served and queued MTs between consecutive epochs. α and β are weighting coefficients (α+β=1) balancing throughput maximization and queue stability. α, being higher, encourages the agent to prioritize service completion, while β forces a stronger penalty on queue growth, ensuring system stability under heavy traffic conditions.
**Algorithm 1.** Pseudo-code of the integrated RL-RQS framework for dynamic queue management.1**Input:** λ: Arrival rate, μ: Service rate, θ: Retry rate, 2      time_sim: Total simulation time, 3      K = Maximum queue length4**Output:** Optimal policy for queue management (based on DQN)5**Define Environment (#1)**6      It contains RQS simulation-based parameters, states, and rewards.7**State Representation (2D vector) (#2)**8-Number of MTs in the queue9-AP status (busy/free)10**Reward Function (#3)**11      R=α·Δ(served clients)−β·Δ(clients in queue)12**Observation (#4)**13-Current queue length14-AP status (busy/free)15**Actions (#5)**16       If Action 1 → Agent does nothing17         If Action 2 → Agent forces AP to serve one MT.

The set reward ensures that the agent continuously maximizes system efficiency without compromising fairness or stability. Over successive training iterations, the agent updates its policy by evaluating the long-term impact of actions rather than short-term gains, ultimately converging toward an optimal service strategy that minimizes latency, reduces queue congestion, and enhances overall throughput. The design of this reward function thus serves as the cornerstone for achieving intelligent, self-adaptive behavior in the proposed 6G queueing model.

Table 3 summarizes the simulation parameters used to evaluate the RL-RQS system. We tried to build a system using a configuration that reflects realistic 6G traffic dynamics while allowing controlled experimentation with different load and congestion levels. We varied the following parameters: (i) the arrival rate (λ) varies across multiple scenarios to emulate different traffic intensities, (ii) the queue size (K) to study its impact on system stability and waiting time, and (iii) the scaling factor (α) and its complementary weight (β=1−α) to govern the trade-off between maximizing service throughput and controlling queue buildup.

We fixed the following parameters: (i) the service rate (μ) to represent the average processing capability of the access point, (ii) the retry rate (θ) to model the frequency of retransmission attempts from orbiting requests, (iii) the simulation time to ensure consistent convergence and performance comparison across all parameter settings, and (iv) the training episodes to enhance the learned knowledge.

The architecture of the DQN agent is shown in Figure 6, which consists of an input layer representing system states that receive a 1×2 feature vector representing the environment state, followed by two fully connected layers with 24 neurons each and ReLU functions, and an output layer estimating Q-values for each possible action. This design enables the system to dynamically adapt to real-time network fluctuations, achieving improved load balancing and reduced waiting time compared to a static queueing approach.

In the next section, we explain the outputs extracted from the simulations and interpret the results. Further investigations are made using first and second-order Markov chain models. The behavior of both spectral gaps described in Section 3.3 allows us to understand deeply the agent’s reward function. Dynamic Time Warping is applied to process the similarity of Markov steady states with different numbers of dimensions.

## 5. Results and Discussion

In this section, we investigate how varying reward weighting influences the learning behavior and the system performance. We present the results of higher-order as well as Markov models and discuss the findings using the DTW metric to validate the results.

### 5.1. Impact of Reward Weighting on RL-Enhanced RQS Performance

Figure 7 and Figure 8 present the performance evaluation of the proposed RL-enhanced RQS model under two different reward configurations, characterized by (K,α,β)=(20,0.1,0.9) and (20,0.9,0.1), respectively. Each experiment spans 10 learning episodes, with 100 time steps per episode, and is conducted across multiple arrival rate scenarios (λ=20,30,50,70,90,100). The subplots illustrate the temporal evolution of served, queued, and orbited MTs, reflecting how the reinforcement learning agent adapts to varying traffic intensities and queue dynamics.

In Figure 7, we plot the served MTs, orbited MTs, and queued MTs for the case (K,α, β)=(20,0.1,0.9). We observe that by setting a high value of the weight β, the reward function places higher emphasis on queue stability rather than maximizing throughput. This initially allowed the agent to adopt a conservative policy, prioritizing queue regulation and minimizing congestion. One can observe, in the first training episodes, that the number of served MTs increases gradually, indicating that the agent is still exploring the environment and refining its understanding of the system’s state transitions. Later, the agent drops in performance, seen as temporary declines in served MTs, which correspond to exploration phases, where the agent tests alternative actions to discover potentially better policies. Over the 10 episodes, the RQS exhibits consistent improvement in serving efficiency, proving that the agent could successfully learn a balanced strategy that stabilizes both the queue and the orbit sizes.

Conversely, Figure 8 (K,α, β)=(20,0.9,0.1) presents the impact of a reward function in prioritizing the throughput maximization. The larger scaling factor α strongly forces the agent to serve as many MTs as possible, and this knowledge gets stronger from episode to episode. As a result, the system experiences a faster rise in the number of served MTs, especially during the first episodes. However, this aggressive serving learned strategy sometimes leads to temporary spikes in the queued MTs, this is obvious because the agent focuses more on short-term service gains rather than maintaining long-term queue balance.

Despite that, the number of queued MTs number eventually decreases as the policy converges, indicating that the agent’s adaptation is in interplay between immediate and cumulative rewards.

Figure 9 presents the distribution of the most frequently selected actions by the RL-enhanced RQS agent across all simulated scenarios. For each neuron output, the action with the maximum Q-weight was identified, and the predominant action across simulations was aggregated and visualized as a histogram. In the figure, red bars represent action 1 “No intervention, the system operates normally” and blue bars represent action 2 “Immediate interaction to force the AP to serve one queued MT”, with the sum of the two always equal to 24 cases for each scenario, reflecting the total number of simulation instances. The output shows that action 1 “No intervention, the system operates normally” is the most chosen action, indicating that the agent has learned to maintain system stability and avoid unnecessary interventions. Action 2 “Immediate interaction to force the AP to serve one queued MT” occurs primarily under high arrival rate conditions (λ = 100), where the server is insufficient to prevent queue accumulation and increased MT orbiting without the agent’s help. One must note that the summation of red and blue bars is exactly 24, indicating the number of cases (6 cased of λ and 4 cases of K). This helps in the comparison between the reward weights.

These observations confirm that the RL agent effectively internalizes system dynamics, selectively intervening only when necessary while allowing natural service progression under moderate or low traffic loads. This demonstrates a learned balance between service efficiency and queue stability, with the agent adapting its policy according to the state of the system. Moreover, the reward weighting factors (α, β), queue size (K), and arrival rate (λ) shape the agent’s decision-making, influencing both the frequency of the interventions and the system’s throughput and queue behavior.

### 5.2. Mixing Time Computation

We analyzed the reward sequences per episode using Markov chain models to capture the convergence of the RL-RQS agent. We started by computing the first- and second-order Markov chains for all 120 simulated cases and for each episode, then we extracted the spectral gaps from the transition matrices. As previously explained, the spectral gaps provide a quantitative measure of the system’s mixing rate. This indicates how quickly the Markov chain reaches its steady state. For each Markov chain, the first and second spectral gaps were computed as represented in Equations (16) and (17), where $\lambda_{2}$ and $\lambda_{3}$ are the second and third largest eigenvalues of the transition matrix, respectively.

The state space $S=\{s_{1},s_{2},\ldots ,s_{n}\}$ represents the reward sequence. The transition tensor $P_{ij}$ captures the probability of moving from state $s_{i}$ to $s_{j}$ in one step. For the second-order Markov chain, $P_{ijk}$ defines transitions considering the previous two states, giving the model the right to capture temporal correlation in the mobile network dynamics. Spectral gaps ${SG}_{1}$ and ${SG}_{2}$ are calculated from the eigenvalues of these transition matrices and quantify global and local mixing rates, respectively. As a result, we can say that they serve as indicators of convergence speed and system stability.

Figure 10 presents the first-order Markov chain (M1) analysis for the simulation case (α,K,λ)=(0.1,10,20). The upper four figures show the transition matrices of the rewards for the first four episodes. We observe that the system exhibits four distinct states in episodes 1 and 2, then it is reduced to two states in episodes 3 and 4. This observation reflects the agent’s policy convergence and stabilization of the reward distribution. The lower four figures depict the corresponding eigenvalues and spectral gaps. The red disk represents the $SG_{1}$ interval (from 1 to $\lambda_{2}$) and the black circle represents $\lambda_{3}$. Notably, $SG_{1}$ increases rapidly in the early episodes as $\lambda_{2}\to 0$, indicating fast convergence toward steady states. In episode 4, a slight reversal occurs, suggesting minor instability or over-exploration by the RL agent. The second spectral gap ($\lambda_{3}$) decreases during the first two episodes and is absent in episodes 3 and 4 due to the reduction to two states, which is consistent with the mathematical expectation for smaller state spaces.

Figure 11 extends this analysis to the second-order Markov chain for the same simulation case. All eight subplots show eigenvalues of the transition matrices for episodes 1–8, with $SG_{1}$ (red disk) and $\lambda_{3}$ (black circle) highlighted. Compared with the first-order chain, the evolution of $SG_{1}$ is slower, reflecting the longer memory inherent in the second-order model. The spectral gap decreases after the 4th episode, indicating the over-learning in the policy.

This slower convergence highlights the ability of the second-order Markov model in capturing temporal dependencies beyond a single episode, providing a more nuanced view of policy dynamics over time.

It is important to mention that monitoring the spectral gaps provides a principled metric to determine the number of episodes necessary for RL training. The findings show that a rapid increase and stabilization of $SG_{1}$ is interpreted by good efficiency of the agent’s policy and that it reached steady-state behavior, while slower or fluctuating spectral gaps suggest continued learning or instability. For this end, we can say that the episode at which $SG_{1}$ converges can be used to identify the minimal number of training episodes required for the agent to achieve a stable and reliable policy. These results meet the 6G smart technologies requirements by providing a quantitative system-specific criterion for RL training duration, avoiding arbitrary choices and ensuring both efficiency and convergence in learning. The 6G smart technologies aim to support ultra-reliable, low-latency network management, this can be achieved by having control algorithms that converge rapidly, maintain stable operation under various dynamic conditions, and efficiently utilize system resources. Therefore, by analyzing the spectral gap of the first- and second-order Markov chains, our RL-RQS agent demonstrates fast convergence to steady-state policies (this is indicated by the rapid stabilization of ${SG}_{1}$) and the ability to keep the reward distribution stable despite variations in network load. The quantitative assessment proves that the agent could easily achieve reliable decision-making within a minimal number of episodes, meeting the 6G requirement for efficient network control without long training or performance fluctuations.

The spectral gap analysis across first- and second-order Markov chains allows us to quantify the number of episodes necessary for the RL agent to achieve a stable policy. Also, the first-order Markov chain offers a rapid estimation of convergence, which can be useful for early-stage evaluation cases. The second-order Markov chain captures longer-term dependencies, enabling the detection of subtle instabilities or over-learning. Combining both orders identifies a practical range of episodes that balances learning efficiency with system stability, guiding the optimal training duration for the RL-enhanced RQS agent. This analysis further clarifies how critical parameters such as reward weighting coefficients directly influence convergence speed, policy stability, and system performance.

### 5.3. Validation and Metrics

To validate the convergence behavior of the RL-enhanced RQS agent and quantify the similarity between the steady-state distributions of different Markov chain orders, we computed the Dynamic Time Warping (DTW) distances of the reward sequences’ steady states for both first- and second-order Markov chains across all episodes and 120 simulated cases. DTW provides a robust metric to compare sequences that may vary in length or temporal alignment, capturing both the rate and pattern of convergence.

The choice of DTW is motivated by its ability to compare sequences that may vary in length or exhibit temporal misalignment, which is common in reward sequences across episodes. Unlike standard probabilistic distance measure methods, DTW captures both the rate and pattern of convergence, providing a more robust metric for the stabilization evaluation of the agent policy over time.

In Figure 12, we present a scatter plot of the DTW distances of the second-order Markov chain versus the first-order Markov chain. Using k-means clustering, three primary clusters emerge. The first cluster converges near zero, indicating cases where the steady states of both Markov chain orders are highly similar, reflecting consistent and rapid convergence of the RL agent’s policy.

The second cluster converges near one, representing the simulation cases where the steady states of the two orders differ significantly; this is likely due to the slower convergence or the presence of longer-term dependencies captured only by the second-order chain. In contrast, the third cluster contains sparse outliers; it reflects the anomalous behavior of the irregular transition of the policy.

Figure 13 shows the histogram of the scatter plot, highlighting the density of DTW distances. The peaks near 0 and 1 confirm the presence of the two dominant convergence patterns: high similarity between the two Markov chain orders for most cases, and significant differences for a smaller set of episodes. The outliers contribute to the low-density tails of the histogram. Markov chain-based analysis validates the convergence assessment. Cases with DTW near zero indicate episodes where the first-order chain is sufficient to capture policy stabilization, whereas DTW near one highlights the episodes where longer memory effects, captured by the second-order chain, are important. This dual-order Markov model DTW comparison serves as a metric for quantifying the DQN-RL convergence and identifying the number of episodes required for further training.

## 6. Conclusions and Future Work

This paper presented an integration framework combining a retrial queueing system in a 6G environment with a deep Q-network reinforcement learning model and Markov-based higher-order analysis to enhance the network performance and achieve a so-called native intelligence technology. Analyzing the first and the second order of the Markov model’s components further brought analytical aspects into system dynamics and reward behavior. By studying the first and second spectral gaps, it was found that they serve as metrics to quantify the convergence speed and capture the optimal number of episodes required for the RL to reach policy stability. The dynamic time warping distance metric was utilized to compare two orders of Markov chain steady-state transitions, revealing the model’s temporal stability. Quantitative results indicate that, with the increase in the training episodes, the served MTs approached 100%, while both queued and orbited MTs decreased to 0%. Across all 120 simulation cases, the most frequently selected action was Action 1 (Do nothing) (approximately 75% of the time), demonstrating the agent’s ability to maintain efficient queue operation without continuously interrupting the server, highlighting the lower energy consumption. These observations reflect the behavior of the system under the simulated cases used in this study. The first- and second-order Markov chain analyses of the agent’s reward sequences confirm rapid convergence, showing that in some scenarios just five episodes were sufficient to achieve stable policy performance, rather than ten episodes. Potential improvements include refining the reward function, incorporating additional state features such as channel quality and mobility patterns, and exploring advanced RL algorithms or multi-agent coordination for denser network scenarios. Limitations of the current study include the reliance on simulation-based evaluation under controlled traffic conditions and the simplified single-agent RL setup.

A continuation step of this work will involve exploring multi-agent coordination, hierarchical learning, and real-world deployment across different conditions to enhance scalability and resilience in ultra-dense network environments.

## Figures and Tables

**Figure 1 sensors-25-07245-f001:**
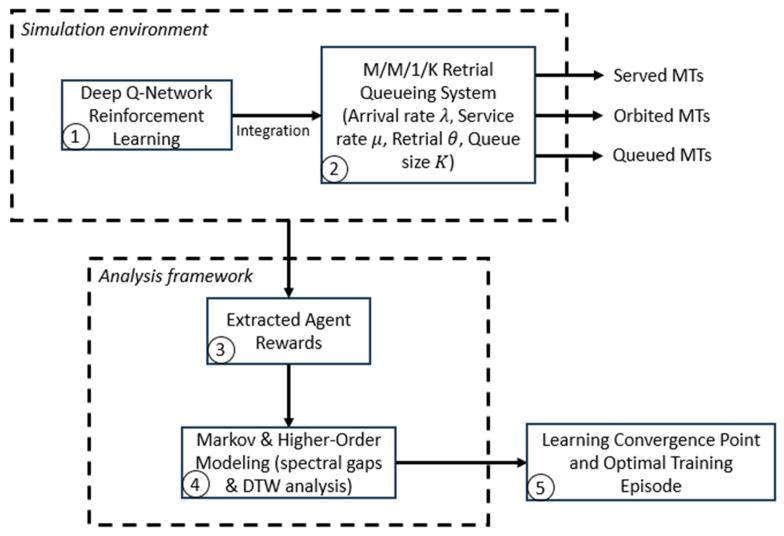
Workflow of the Retrial Queueing System Integrated with DQN-RL and Markov Modeling.

**Figure 2 sensors-25-07245-f002:**
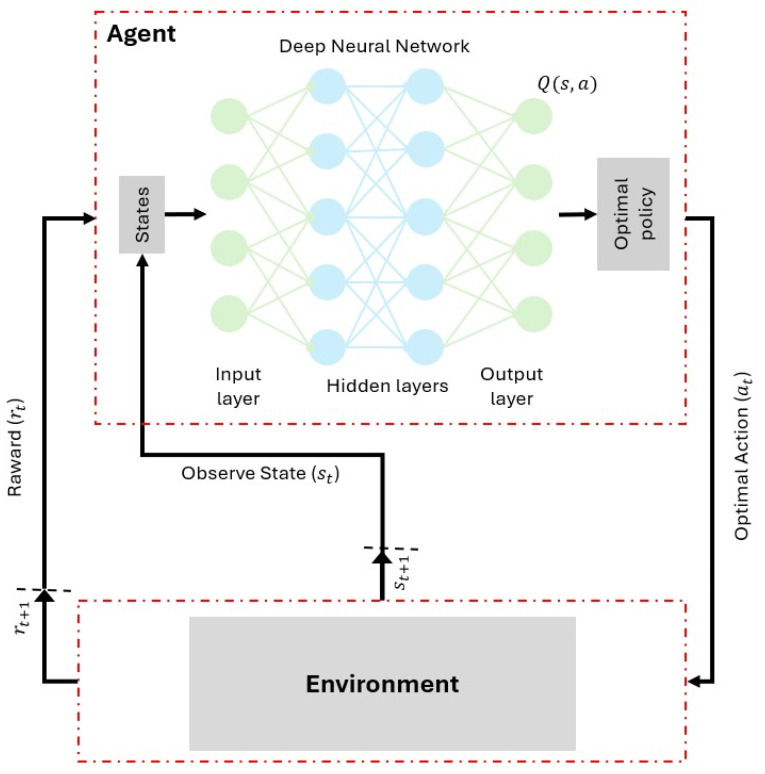
Structure of the DQN Framework for Reinforcement Learning.

**Figure 3 sensors-25-07245-f003:**
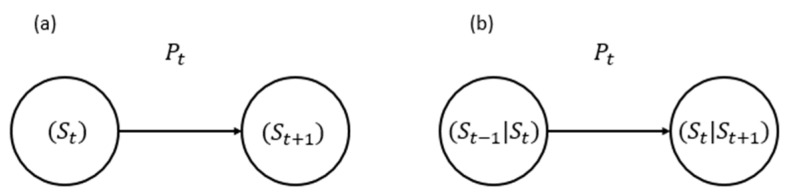
State Transition Representation in: (**a**) first-order and (**b**) second-order Markov models.

**Figure 4 sensors-25-07245-f004:**
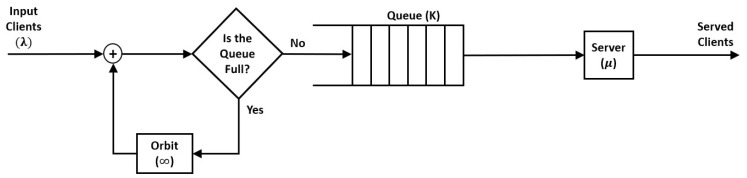
The Retrial Queueing System model structure represents the 6G communication access process with arrival, service, and retrial mechanisms.

**Figure 5 sensors-25-07245-f005:**
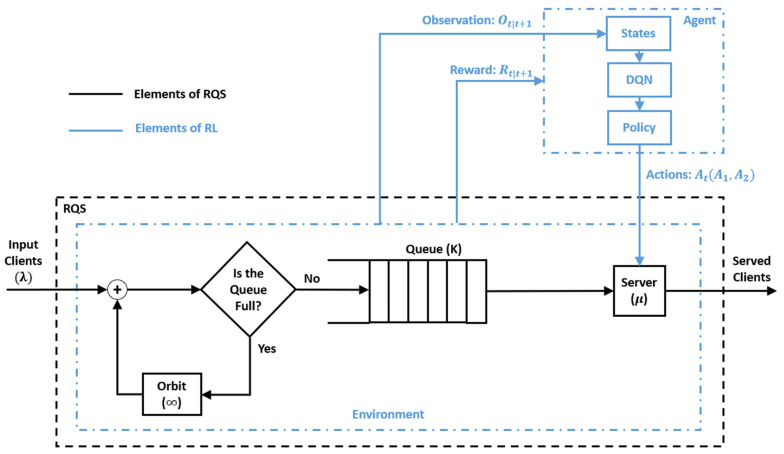
Integration of the DQN-RL framework within the RQS model.

**Figure 6 sensors-25-07245-f006:**
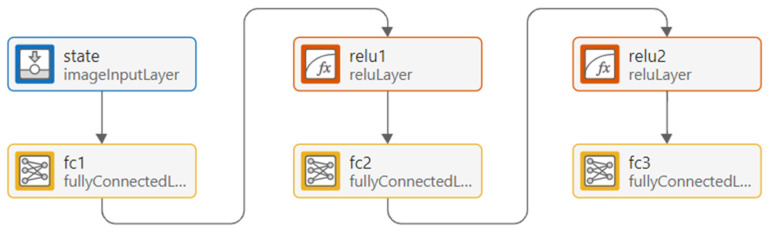
Internal neural network architecture of the DQN agent used for Q-value approximation.

**Figure 7 sensors-25-07245-f007:**
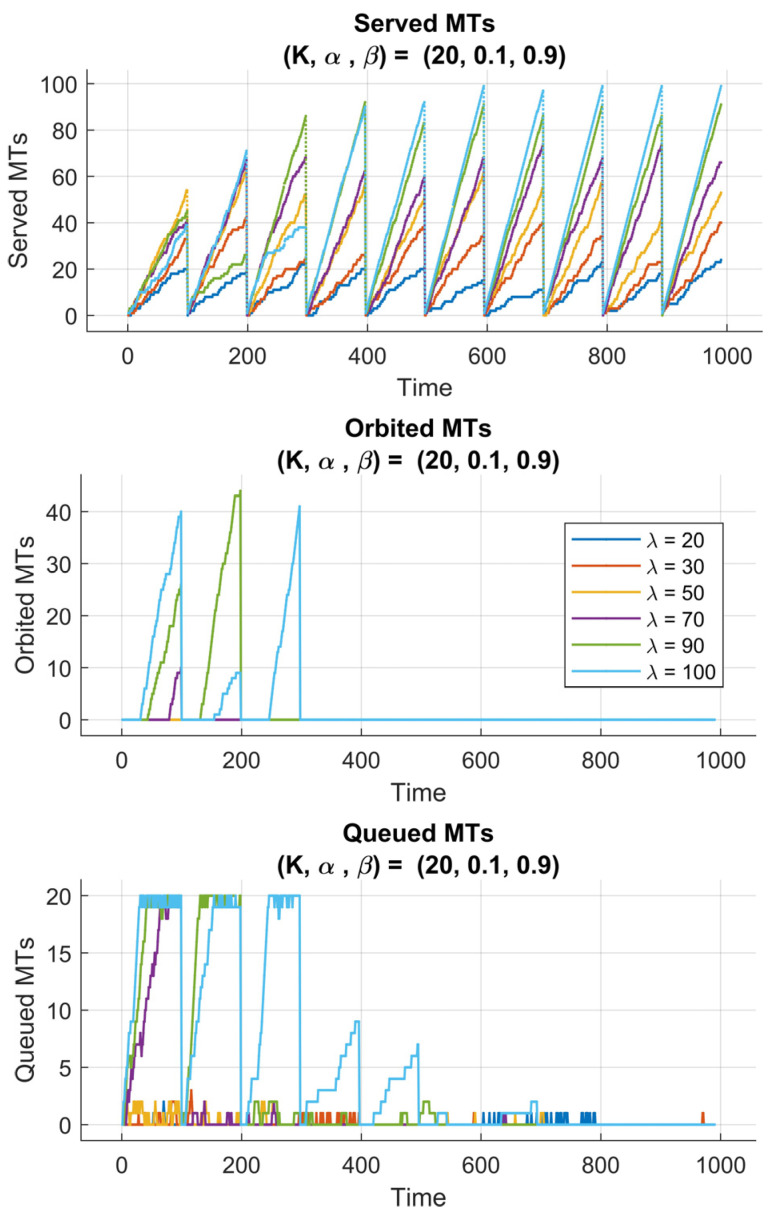
Performance of 6G-RQS with RL Integration for (K,α,β) = (20,0.1,0.9).

**Figure 8 sensors-25-07245-f008:**
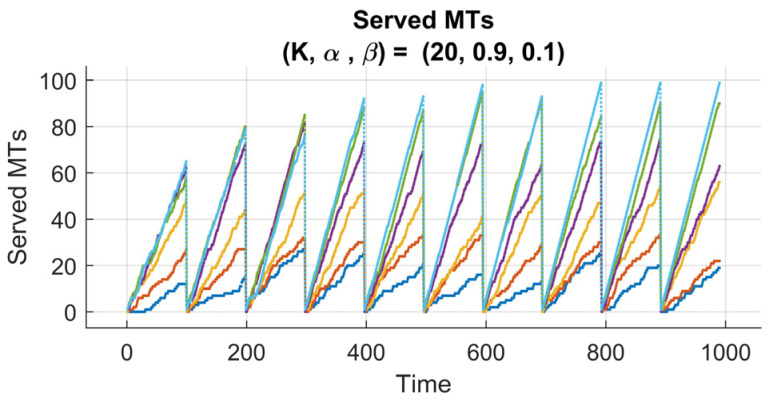
Performance of 6G-RQS with RL Integration for (K,α,β) = (20,0.9,0.1).

**Figure 9 sensors-25-07245-f009:**
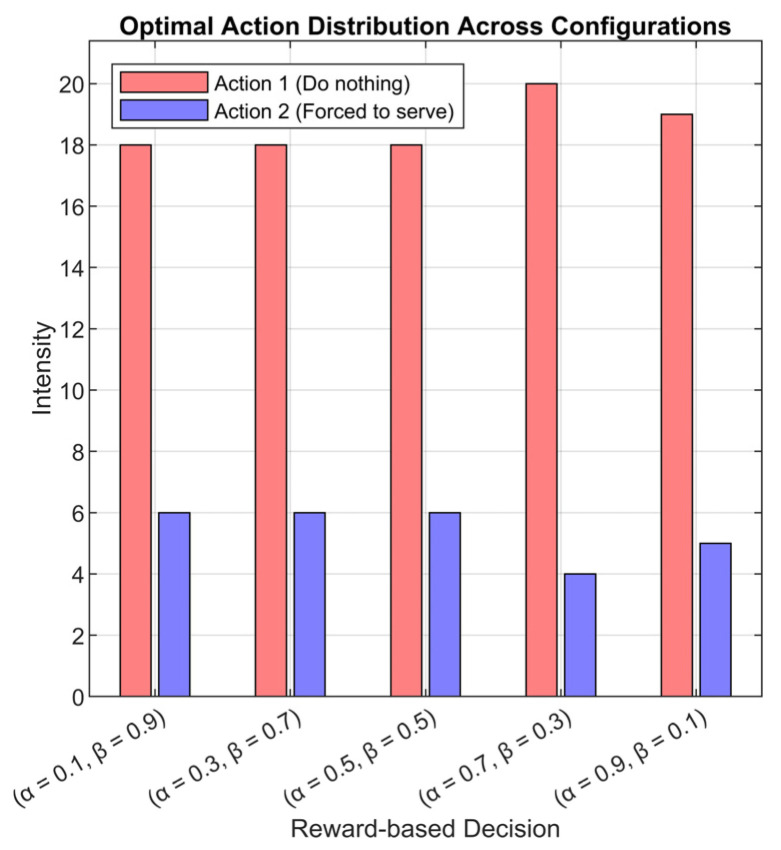
Histogram of the most frequently selected actions by the RL agent across all simulation cases.

**Figure 10 sensors-25-07245-f010:**
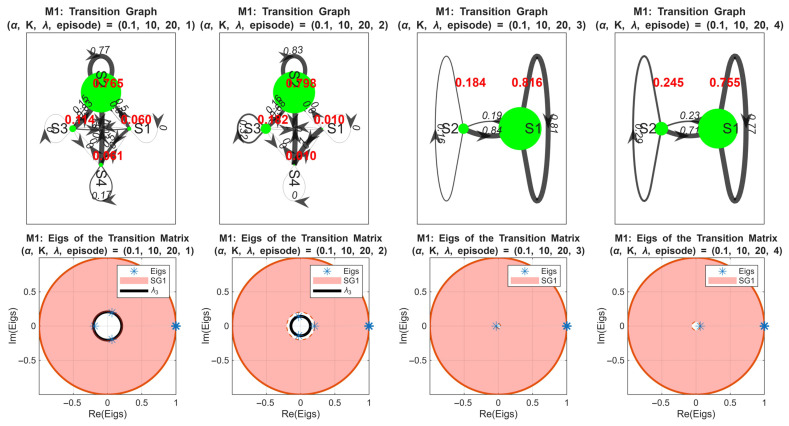
First-order Markov chain analysis of rewards for $\left(\alpha ,K,\lambda \right)=\left(0.1,10,20\right)$.

**Figure 11 sensors-25-07245-f011:**
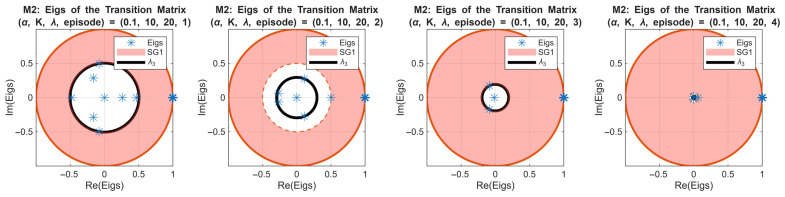
Second-order Markov chain eigenvalues and spectral gaps across episodes 1–8 for $\left(\alpha ,K,\lambda \right)=\left(0.1,10,20\right)$.

**Figure 12 sensors-25-07245-f012:**
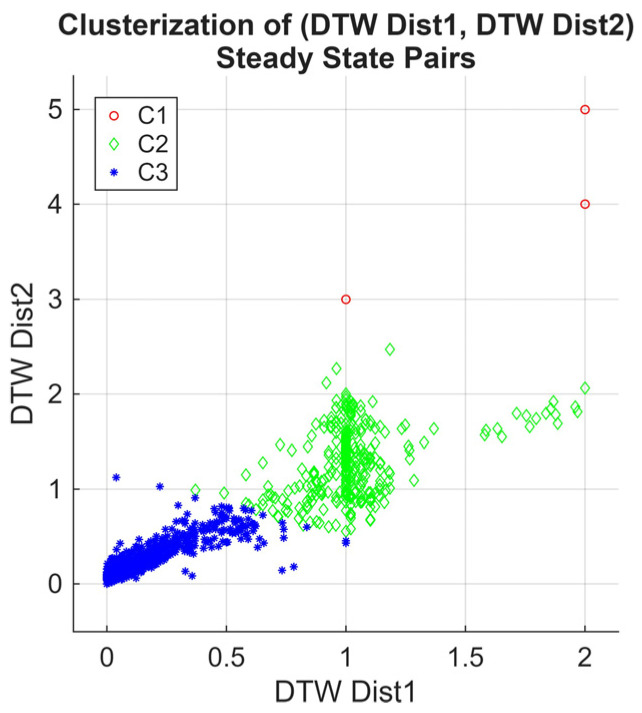
Scatter plot of DTW distances between second-order and first-order Markov chain steady states across all episodes and simulation cases, with k-means clustering highlighting three convergence patterns.

**Figure 13 sensors-25-07245-f013:**
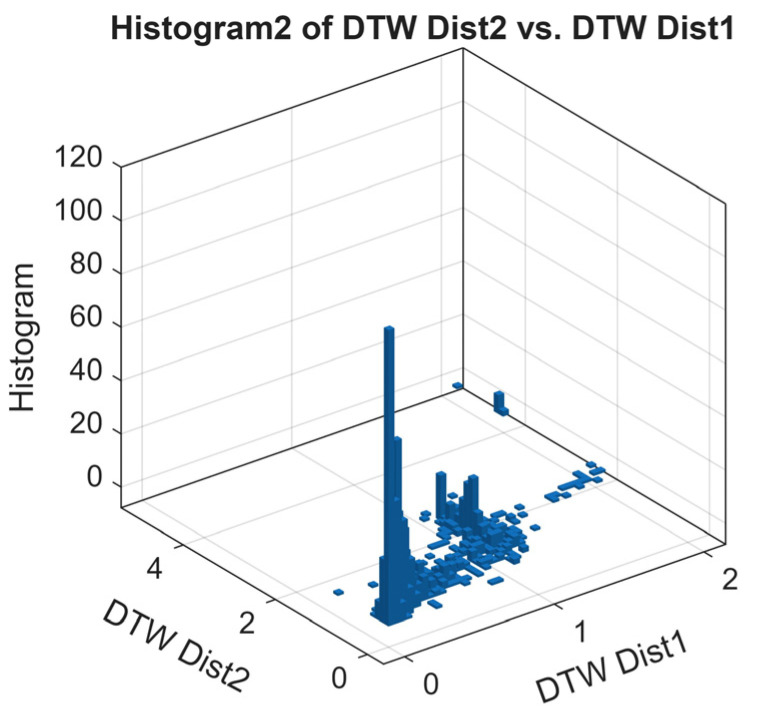
Histogram of DTW distances from Figure 12, illustrating the distribution and density of cases converging near 0 and 1, along with outlier episodes.

**Table 1 sensors-25-07245-t001:** RL-DQN key components.

DQN Key Components	Explanation
State Encoding	System information is mapped to a feature vector and used as the network input.
Action Selection	Actions are chosen according to an ε−greedy policy, ensuring a balance between exploration of new strategies and exploitation of learned policies.
Reward Mechanism	A function used to quantify the impact of the chosen action on system performance, incorporating metrics such as the KPIs.
Experience Reply	Interaction samples (s, a, r, s′) are stored in a reply buffer and randomly sampled during training, which reduces temporal correlations and improves convergence stability.
Target Network	Two neural networks are maintained: the online network with parameter θ and a periodically updated target network with parameter $\theta^{-}$. This separation mitigates stability during Q-value updates.

**Table 2 sensors-25-07245-t002:** Value ranges and interpretation of the first- and second-order spectral gaps (${SG}_{1},{SG}_{2}$).

${SG}_{1},{SG}_{2}$ Range	Qualitative Meaning	Dynamic Interpretation
${SG}_{1}\;\approx \;0,\;\;{SG}_{2}\;\approx \;0$	Both near zero	Chain is almost reducible; extremely slow mixing; possibly multiple disconnected or nearly isolated regions.
${SG}_{1}$ small, ${SG}_{2}\;\gg \;{SG}_{1}$	Large separation	One dominant bottleneck, the system has two large metastable basins; after crossing the bottleneck, everything else mixes fast.
${SG}_{1}\;\approx \;{SG}_{2}$, both moderate	Several slow modes	Multiscale metastability: multiple regions with similar sluggish exchange rates; several comparable timescales.
SG_1_ moderate, SG_2_ ≈ 1	Fast local equilibration after the main bottleneck	Single dominant slow process, but local structure is quickly equilibrating a “clean” two-phase system.
SG_1_ ≈ 1, SG_2_ ≈ 1	Both large	Chain mixes very rapidly; nearly uniform transition probabilities; no metastable structure.

**Table 3 sensors-25-07245-t003:** Used parameter for the simulation.

Parameter	Type	Value
Arrival rate (λ)	Variable	[20,30,50,70,90,100]
Service rate (μ)	Fixed	0.5
Retry rate (θ)	Fixed	1
Queue size (K)	Variable	[10,15,20,40]
Scaling factor (α)	Variable	[0.1, 0.3, 0.5, 0.7, 0.9]
Weight (β)	Variable	1−α
Simulation time	Fixed	100 units
Number of Episodes	Fixed	10

## Data Availability

The data were generated using the algorithms outlined in the article.

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
