# Peer review of "Higher-Order Markov Model-Based Analysis of Reinforcement Learning in 6G Mobile Retrial Queueing Systems"

_sensors, 2025, doi:10.3390/s25237245_

Round 1
Reviewer 1 Report
Comments and Suggestions for Authors
The paper proposes an integrated queuing system based on 6G technology, adopting a method that combines deep Q-network reinforcement learning with a re-examination queuing system to provide a new approach for adaptive queue management and service optimization, whose performance has been verified through relevant experiments. Suggestions for the paper are as follows:
- In Section 1.2 of the paper, regarding content such as "Markovian queueing systems" and "Reinforcement Learning (RL) based optimization for network management" carried out in References 2, 3, and 4, it is recommended to provide detailed descriptions and add a comparative analysis between the paper's work and these studies.
- For the description of the loss function in Formula 8, some symbols have not been explained, and supplementary explanations are suggested.
- The meanings of some symbols appearing in Figures 4 and 5 are not labeled, so please add them.
- In Algorithm 1 "Pseudo-code of the integrated RL-RQS framework for dynamic queue management", is the maximum queue length "k" in the third line the same as the queue size "K" in Table 3? If so, it is recommended to unify the format. "Action 1/2" is mentioned in Lines 284 and 285 of the paper, but it is described as "Action = 0/1" in the last two lines of Algorithm 1; please unify the format.
- In the neural network architecture of Figure 6, the input layer is labeled as "imageInputLayer"—is there an error here? Are the input values of the neural network used in the paper image data? Additionally, it is recommended to add a specific description of the neural network architecture in the figure caption, such as the specific dimensions of the input layer, hidden layers, and output layer.
- Line 336 of the paper mentions that a high weight value is used for the scaling factor, but the actual setting in the paper is 0.1, which is not a high weight value—please explain this inconsistency.
- Line 413 of the paper states "These results meet the 6G smart technologies requirements", but the paper does not mention the specific requirements of 6G smart technologies for the experiment, nor does the experimental result analysis section explain how the requirements are specifically met.
- In the final conclusion section of the paper, it is recommended to conduct a targeted summary based on the three research gaps proposed by the authors: fixed parameters, limitations of RL application scenarios, and the lack of integration between retrial queuing and RL. This will illustrate that the proposed problems have been solved and strengthen the logical closure of the paper.
Reviewer 2 Report
Comments and Suggestions for Authors
See the attach.file

Reviewer 3 Report
Comments and Suggestions for Authors
- The introduction gives an overview of the challenges of 6G, but it does not clearly explain the specific research gap about combining retrial queueing systems with reinforcement learning.
- In the research, only 10 training episodes were used, which is not enough for the DQN to achieve a stable result. It may lead to results that are not statistically significant.
- There is no comparison to measure performance improvement, such as against classical Q-learning or a queueing system without reinforcement learning.
- Using Dynamic Time Warping (DTW) to compare steady-state distributions needs a good reason. Why not just use regular probabilistic distance measures?
- No tables are showing key performance metrics, such as throughput, latency, or average queue size.
- Figures 7–13 are conceptually informative but lack clear axis labels, units, and legends.
- The claim that the study “provides a strong basis for improving 6G queueing strategies” is not supported.
- The conclusions can be adapted, given the limited experimental evidence.
The English language is mostly clear, but needs some editing. There are several grammatical and spelling mistakes, and some technical terms are used incorrectly.
Round 2
Reviewer 3 Report
Comments and Suggestions for Authors
- It would be helpful to add a clear statement about the scope in the Abstract and Introduction: the focus is on convergence (policy stabilization, spectral gap, minimal training), while benchmarking is outside the scope of this paper.
- In the conclusion or abstract, soften the claims "enhance performance/low energy consumption" to indicate they are based on simulation conditions (without benchmarking).
Acceptable
